# Non-monotone Submodular Maximization in Exponentially Fewer Iterations

**Eric Balkanski**
Harvard University
`ericbalkanski@g.harvard.edu`

**Adam Breuer**
Harvard University
`breuer@g.harvard.edu`

**Yaron Singer**
Harvard University
`yaron@seas.harvard.edu`

## Abstract

In this paper we consider parallelization for applications whose objective can be expressed as maximizing a non-monotone submodular function under a cardinality constraint. Our main result is an algorithm whose approximation is arbitrarily close to $1/2e$ in $\mathcal{O}(\log^2 n)$ adaptive rounds, where $n$ is the size of the ground set. This is an exponential speedup in parallel running time over any previously studied algorithm for constrained non-monotone submodular maximization. Beyond its provable guarantees, the algorithm performs well in practice. Specifically, experiments on traffic monitoring and personalized data summarization applications show that the algorithm finds solutions whose values are competitive with state-of-the-art algorithms while running in exponentially fewer parallel iterations.

## 1 Introduction

In machine learning, many fundamental quantities we care to optimize such as entropy, graph cuts, diversity, coverage, diffusion, and clustering are submodular functions. Although there has been a great deal of work in machine learning on applications that require constrained *monotone* submodular maximization, many interesting submodular objectives are non-monotone. Constrained non-monotone submodular maximization is used in large-scale personalized data summarization applications such as image summarization, movie recommendation, and revenue maximization in social networks [MBK16]. In addition, many data mining applications on networks require solving constrained max-cut problems (see Section 4).

Non-monotone submodular maximization is well-studied [FMV11, LMNS09, GRST10, FNS11, GV11, BFNS14, CJV15, MBK16, EN16], particularly under a cardinality constraint [LMNS09, GRST10, GV11, BFNS14, MBK16]. For maximizing a non-monotone submodular function under a cardinality constraint $k$, a simple randomized greedy algorithm that iteratively includes a random element from the set of $k$ elements with largest marginal contribution at every iteration achieves a $1/e$ approximation to the optimal set of size $k$ [BFNS14]. For more general constraints, Mirzasoleiman et al. develop an algorithm with strong approximation guarantees that works well in practice [MBK16].

While the algorithms for constrained non-monotone submodular maximization achieve strong approximation guarantees, their parallel runtime is linear in the size of the data due to their high *adaptivity*. Informally, the adaptivity of an algorithm is the number of sequential rounds it requires when polynomially-many function evaluations can be executed in parallel in each round. The adaptivity of the randomized greedy algorithm is $k$ since it sequentially adds elements in $k$ rounds. The algorithm in Mirzasoleiman et al. is also $k$-adaptive, as is any known constant approximation

algorithm for constrained non-monotone submodular maximization. In general, $k$ may be $\Omega(n)$, and hence the adaptivity as well as the parallel runtime of all known constant approximation algorithms for constrained submodular maximization are at least *linear* in the size of the data.

For large-scale applications we seek algorithms with *low* adaptivity. Low adaptivity is what enables algorithms to be efficiently parallelized (see Appendix A for further discussion). For this reason, adaptivity is studied across a wide variety of areas including online learning [NSYD17], ranking [Val75, Col88, BMW16], multi-armed bandits [AAAK17], sparse recovery [HNC09, IPW11, HBCN09], learning theory [CG17, BGSMdW12, CST+17], and communication complexity [PS84, DGS84, NW91]. For submodular maximization, somewhat surprisingly, until very recently $\Omega(n)$ was the best known adaptivity (and hence best parallel running time) required for a constant factor approximation to monotone submodular maximization under a cardinality constraint. Although there has been a great deal of work on *distributed* submodular optimization (e.g. in the Map-Reduce model), the algorithms for distributed submodular optimization address the challenges associated with processing data that exceeds memory capacity. These algorithms partition the ground set to multiple machines and run sequential greedy algorithms on each machine separately and are therefore $\Omega(n)$-adaptive in the worst case (e.g. [CKT10, KMVV15, MKSK13, MZ15, BENW16]).

A recent line of work introduces new techniques for maximizing *monotone* submodular functions under a cardinality constraint that produce algorithms that are $\mathcal{O}(\log n)$-adaptive and achieve both strong constant factor approximation guarantees [BS18a, BS18b] and even optimal approximation guarantees [BRS18, EN18]. This is tight in the sense that no algorithm can achieve a constant factor approximation with $\tilde{o}(\log n)$ rounds [BS18a]. Unfortunately, these techniques are only applicable to monotone submodular maximization and can be arbitrarily bad in the non-monotone case.

*Is it possible to design fast parallel algorithms for non-monotone submodular maximization?*

For unconstrained non-monotone submodular maximization, one can trivially obtain an approximation of $1/4$ in 0 rounds by simply selecting a set uniformly at random [FMV11]. We therefore focus on the problem of maximizing a non-monotone submodular function under a cardinality constraint.

**Main result.** Our main result is the BLITS algorithm, which obtains an approximation ratio arbitrarily close to $1/2e$ for maximizing a non-monotone (or monotone) submodular function under a cardinality constraint in $\mathcal{O}(\log^2 n)$ adaptive rounds (and $\mathcal{O}(\log^3 n)$ parallel runtime — see Appendix A), where $n$ is the size of the ground set. Although its approximation ratio is about half of the best known approximation for this problem [BFNS14], it achieves its guarantee in exponentially fewer rounds. Furthermore, we observe across a variety of experiments that despite this slightly weaker worst-case approximation guarantee, BLITS consistently returns solutions that are competitive with the state-of-the-art, and does so exponentially faster.

**Technical overview.** Non-monotone submodular functions are notoriously challenging to optimize. Unlike in the monotone case, standard algorithms for submodular maximization such as the greedy algorithm perform arbitrarily poorly on non-monotone functions, and the best achievable approximation remains unknown.[1] Since the marginal contribution of an element to a set is not guaranteed to be non-negative, an algorithm's local decisions in the early stages of optimization may contribute negatively to the value of its final solution. At a high level, we overcome this problem with an algorithmic approach that iteratively adds to the solution blocks of elements obtained after aggressively discarding other elements. Showing the guarantees for this algorithm on non-monotone functions requires multiple subtle components. Specifically, we require that at every iteration, any element is added to the solution with low probability. This requirement imposes a significant additional challenge to just finding a block of high contribution at every iteration, but it is needed to show that in future iterations there will exist a block with large contribution to the solution. Second, we introduce a pre-processing step that discards elements with negative expected marginal contribution to a random set drawn from some distribution. This pre-processing step is needed for two different arguments: the first is that a large number of elements are discarded at every iteration, and the second is that a random block has high value when there are $k$ surviving elements.

**Paper organization.** Following a few preliminaries, we present the algorithm and its analysis in sections 2 and 3. We present the experiments in Section 4.

**Preliminaries.** A function $f : 2^N \to \mathbb{R}_+$ is *submodular* if the marginal contributions $f_S(a) := f(S \cup a) - f(S)$ of an element $a \in N$ to a set $S \subseteq N$ are diminishing, i.e., $f_S(a) \geq f_T(a)$ for all $a \in N \setminus T$ and $S \subseteq T$. It is monotone if $f(S) \leq f(T)$ for all $S \subseteq T$. We assume that $f$ is non-negative, i.e., $f(S) \geq 0$ for all $S \subseteq N$, which is standard. We denote the optimal solution by $O$, i.e. $O := \operatorname{argmax}_{|S| \leq k} f(S)$, and its value by $\texttt{OPT} := f(O)$. We use the following lemma from [FMV11], which is useful for non-monotone functions:

**Lemma 1** ([FMV11])**.** *Let $g : 2^N \to \mathbb{R}$ be a non-negative submodular function. Denote by $A(p)$ a random subset of $A$ where each element appears with probability at most $p$ (not necessarily independently). Then, $\mathbb{E}\left[g(A(p))\right] \geq (1-p)g(\emptyset) + p \cdot g(A) \geq (1-p)g(\emptyset)$.*

**Adaptivity.** Informally, the adaptivity of an algorithm is the number of sequential rounds it requires when polynomially-many function evaluations can be executed in parallel in each round. Formally, given a function $f$, an algorithm is $r$-adaptive if every query $f(S)$ for the value of a set $S$ occurs at a round $i \in [r]$ such that $S$ is independent of the values $f(S')$ of all other queries at round $i$.

## 2 The BLITS Algorithm

In this section, we describe the BLock ITeration Submodular maximization algorithm (henceforth BLITS), which obtains an approximation arbitrarily close to $1/2e$ in $\mathcal{O}(\log^2 n)$ adaptive rounds. BLITS iteratively identifies a block of at most $k/r$ elements using a SIEVE subroutine, treated as a black-box in this section, and adds this block to the current solution $S$, for $r$ iterations.

---
**Algorithm 1** BLITS: the BLock ITeration Submodular maximization algorithm
---
**Input:** constraint $k$, bound on number of iterations $r$
  $S \leftarrow \emptyset$
  **for** $r$ iterations $i = 1$ to $r$ **do**
    $S \leftarrow S \cup \textsc{Sieve}(S, k, i, r)$
  **return** $S$

---

The main challenge is to find in logarithmically many rounds a block of size at most $k/r$ to add to the current solution $S$. Before describing and analyzing the SIEVE subroutine, in the following lemma we reduce the problem of showing that BLITS obtains a solution of value $\alpha v^\star / e$ to showing that SIEVE finds a block with marginal contribution at least $(\alpha/r)((1 - 1/r)^{i-1} v^\star - f(S_{i-1}))$ to $S$ at every iteration $i$, where we wish to obtain $v^\star$ close to $\texttt{OPT}$. The proof generalizes an argument in [BFNS14] and is deferred to Appendix B.

**Lemma 2.** *For any $\alpha \in [0,1]$, assume that at iteration $i$ with current solution $S_{i-1}$, SIEVE returns a random set $T_i$ s.t. $\mathbb{E}\left[f_{S_{i-1}}(T_i)\right] \geq \frac{\alpha}{r}\left(\left(1 - \frac{1}{r}\right)^{i-1} v^\star - f(S_{i-1})\right)$. Then, $\mathbb{E}\left[f(S_r)\right] \geq \frac{\alpha}{e} \cdot v^\star$.*

The advantage of BLITS is that it terminates after $\mathcal{O}(d \cdot \log n)$ adaptive rounds when using $r = \mathcal{O}(\log n)$ and a SIEVE subroutine that is $d$-adaptive. In the next section we describe SIEVE and prove that it respects the conditions of Lemma 2 in $d = \mathcal{O}(\log n)$ rounds.

## 3 The SIEVE Subroutine

In this section, we describe and analyze the SIEVE subroutine. We show that for any constant $\epsilon > 0$, this algorithm finds in $\mathcal{O}(\log n)$ rounds a block of at most $k/r$ elements with marginal contribution to $S$ that is at least $t/r$, with $t := ((1 - \epsilon/2)/2)((1 - 1/r)^{i-1}(1 - \epsilon/2)\texttt{OPT} - f(S_{i-1}))$, when called at iteration $i$ of BLITS. By Lemma 2 with $\alpha = (1 - \epsilon)/2$ and $v^\star = (1 - \epsilon/2)\texttt{OPT}$, this implies that BLITS obtains an approximation arbitrarily close to $1/2e$ in $\mathcal{O}(\log^2 n)$ rounds.

The SIEVE algorithm, described formally below, iteratively discards elements from a set $X$ initialized to the ground set $N$. We denote by $\mathcal{U}(X)$ the uniform distribution over all subsets of $X$ of size

exactly $k/r$ and by $\Delta(a, S, X)$ the expected marginal contribution of an element $a$ to a union of the current solution $S$ and a random set $R \sim \mathcal{U}(X)$, i.e.

$$\Delta(a, S, X) := \mathbb{E}_{R \sim \mathcal{U}(X)} \left[ f_{S \cup (R \setminus a)}(a) \right].$$

At every iteration, SIEVE first pre-processes surviving elements $X$ to obtain $X^+$, which is the set of elements $a \in X$ with non-negative marginal contribution $\Delta(a, S, X)$. After this pre-processing step, SIEVE evaluates the marginal contribution $\mathbb{E}_{R \sim \mathcal{U}(X)}[f_S(R \cap X^+)]$ of a random set $R \sim \mathcal{U}(X)$ without its elements not in $X^+$ (i.e. $R$ excluding its elements with negative expected marginal contribution). If the marginal contribution of $R \cap X^+$ is at least $t/r$, then $R \cap X^+$ is returned. Otherwise, the algorithm discards from $X$ the elements $a$ with expected marginal contribution $\Delta(a, S, X)$ less than $(1 + \epsilon/2)t/k$. The algorithm iterates until either $\mathbb{E}[f_S(R \cap X^+)] \geq t/r$ or there are less than $k$ surviving elements, in which case SIEVE returns a random set $R \cap X^+$ with $R \sim \mathcal{U}(X)$ and with dummy elements added to $X$ so that $|X| = k$. A dummy element $a$ is an element with $f_S(a) = 0$ for all $S$.

---

**Algorithm 2** SIEVE$(S, k, i, r)$

---

**Input:** current solution $S$ at outer-iteration $i \leq r$

$\quad X \leftarrow N, t \leftarrow \frac{1-\epsilon/2}{2}((1 - 1/r)^{i-1}(1 - \epsilon/2)\texttt{OPT} - f(S))$
$\quad$**while** $|X| > k$ **do**
$\quad\quad X^+ \leftarrow \{a \in X \ : \ \Delta(a, S, X) \geq 0\}$
$\quad\quad$**if** $\mathbb{E}_{R \sim \mathcal{U}(X)}[f_S(R \cap X^+)] \geq t/r$ **return** $R \cap X^+$, where $R \sim \mathcal{U}(X)$
$\quad\quad X \leftarrow \{a \in X \ : \ \Delta(a, S, X) \geq (1 + \epsilon/4)\,t/k\}$
$\quad X \leftarrow X \cup \{k - |X| \text{ dummy elements}\}$
$\quad X^+ \leftarrow \{a \in X \ : \ \Delta(a, S, X) \geq 0\}$
$\quad$**return** $R \cap X^+$, where $R \sim \mathcal{U}(X)$

---

The above description is an idealized version of the algorithm. In practice, we do not know $\texttt{OPT}$ and we cannot compute expectations exactly. Fortunately, we can apply multiple guesses for $\texttt{OPT}$ non-adaptively and obtain arbitrarily good estimates of the expectations in one round by sampling. The sampling process for the estimates first samples $m$ sets from $\mathcal{U}(X)$, then queries the desired sets to obtain a random realization of $f_S(R \cap X^+)$ and $f_{S \cup (R \setminus a)}(a)$, and finally averages the $m$ random realizations of these values. By standard concentration bounds, $m = \mathcal{O}((\texttt{OPT}/\epsilon)^2 \log(1/\delta))$ samples are sufficient to obtain with probability $1 - \delta$ an estimate with an $\epsilon$ error. For ease of presentation and notation, we analyze the idealized version of the algorithm, which easily extends to the algorithm with estimates and guesses as in [BS18a, BS18b, BRS18]. Due to lack of space, we only include proof sketches for some lemmas and defer full proofs to Appendix D.

### 3.1 The approximation

Our goal is to show that SIEVE returns a random block whose expected marginal contribution to $S$ is at least $t/r$. By Lemma 2 this implies BLITS obtains a $(1 - \epsilon)/2e$-approximation.

**Lemma 3.** *Assume $r \geq 20\rho\epsilon^{-1}$ and that after at most $\rho - 1$ iterations of* SIEVE, SIEVE *returns a set $R$ at iteration $i$ of* BLITS, *then* $\mathbb{E}[f_S(R)] \geq \frac{t}{r} = \frac{1-\epsilon/2}{2r}\left((1 - \frac{1}{r})^{i-1}(1 - \epsilon/2)\texttt{OPT} - f(S)\right)$.

The remainder of the analysis of the approximation is devoted to the proof of Lemma 3. First note that if SIEVE returns $R \cap X^+$, then the desired bound on $\mathbb{E}[f_S(R)]$ follows immediately from the condition to return that block. Otherwise SIEVE returns $R$ due to $|X| \leq k$, and then the proof consists of two parts. First, in Section 3.1.1 we argue that when SIEVE terminates, there *exists* a subset $T$ of $X$ for which $f_S(T) \geq t$. Then, in Section 3.1.2 we prove that such a subset $T$ of $X$ for which $f_S(T) \geq t$ not only exists, but is also returned by SIEVE. We do this by proving a new general lemma for non-monotone submodular functions that may be of independent interest. This lemma shows that a random subset of $X$ of size $s$ well approximates the optimal subset of size $s$ in $X$.

#### 3.1.1 Existence of a surviving block with high contribution to $S$

The main result in this section is Lemma 6, which shows that when SIEVE terminates there *exists* a subset $T$ of $X$ s.t $f_S(T) \geq t$. To prove this, we first prove Lemma 4, which argues that $f(O \cup S) \geq$

$(1-1/r)^{i-1}\mathtt{OPT}$. This bound explains the $(1-1/r)^{i-1}(1-\epsilon/2)\mathtt{OPT} - f(S_{i-1}))$ term in $t$. For monotone functions, this is trivial since $f(O \cup S) \geq f(O) = \mathtt{OPT}$ by definition of monotonicity. For non-monotone functions, this inequality does not hold. Instead, the approach used to bound $f(O \cup S)$ is to argue that any element $a \in N$ is added to $S$ by SIEVE with probability at most $1/r$ at every iteration. The key to that argument is that in both cases where SIEVE terminates we have $|X| \geq k$ (with $X$ possibly containing dummy elements), which implies that every element $a$ is in $R \sim \mathcal{U}(X)$ with probability at most $1/r$.

**Lemma 4.** *Let $S$ be the set obtained after $i-1$ iterations of* BLITS *calling the* SIEVE *subroutine, then* $\mathbb{E}[f(O \cup S)] \geq (1-1/r)^{i-1}\mathit{OPT}$.

*Proof.* In both cases where SIEVE terminates, $|X| \geq k$. Thus $\Pr[a \in R \sim \mathcal{U}(X)] = k/(r|X|) < 1/r$. This implies that at iteration $i$ of BLITS, $\Pr[a \in S] \leq 1 - (1-1/r)^{i-1}$. Next, we define $g(T) := f(O \cup T)$, which is also submodular. By Lemma 1 from the preliminaries, we get

$$\mathbb{E}[f(S \cup O)] = \mathbb{E}[g(S)] \geq (1-1/r)^{i-1}g(\emptyset) = (1-1/r)^{i-1}\mathtt{OPT}. \qquad \square$$

Let $\rho$, $X_j$, and $R_j$ denote the number of iterations of SIEVE$(S, k, i, r)$, the set $X$ at iteration $j \leq \rho$ of SIEVE, and the set $R \sim \mathcal{U}(X_j)$ respectively. We show that the expected marginal contribution of $O$ to $S \cup \left(\cup_{j=1}^{\rho}R_j\right)$ approximates $(1-1/r)^{i-1}\mathtt{OPT} - f(S)$ well. This crucial fact allows us to argue about the value of optimal elements that survive iterations of SIEVE. We defer the proof to Appendix C.

**Lemma 5.** *For all $r, \rho, \epsilon > 0$ s.t. $r \geq 20\rho\epsilon^{-1}$, if* SIEVE$(S, k, i, r)$ *has not terminated after $\rho$ iterations, then* $\mathbb{E}_{R_1,\ldots,R_\rho}\left[f_{S\cup\left(\cup_{j=1}^{\rho}R_j\right)}(O)\right] \geq (1-\epsilon/10)\left((1-1/r)^{i-1}(1-\epsilon/2)\mathit{OPT} - f(S)\right).$

We are now ready to show that when SIEVE terminates after $\rho$ iterations, there exists a subset $T$ of $X_\rho$ s.t $f_S(T) \geq t$. At a high level, the proof defines $T$ to be a set of meaningful optimal elements, then uses Lemma 5 to show that these elements survive $\rho$ iterations of SIEVE and respect $f_S(T) \geq t$.

**Lemma 6.** *For all $r, \rho, \epsilon > 0$, if $r \geq 20\rho\epsilon^{-1}$, then there exists $T \subseteq X_\rho$, that survives $\rho$ iterations of* SIEVE$(S, k, i, r)$ *and that satisfies* $f_S(T) \geq \frac{1-\epsilon/10}{2}\left((1-1/r)^{i-1}(1-\epsilon/2)\mathit{OPT} - f(S)\right).$

*Proof Sketch; full proof in Appendix C.* Let $O = \{o_1, \ldots, o_k\}$ be the optimal elements in some arbitrary order and $O_\ell = \{o_1, \ldots, o_\ell\}$. We define $\Delta_\ell := \mathbb{E}_{R_1,\ldots,R_\rho}[f_{S\cup O_{\ell-1}\cup\left(\cup_{j=1}^{\rho}R_j\setminus\{o_\ell\}\right)}(o_\ell)]$ and $T$ to be the set of optimal elements $o_\ell$ such that $\Delta_\ell \geq \frac{1}{2k} \cdot \mathbb{E}_{R_1,\ldots,R_\rho}[f_{S\cup\left(\cup_{j=1}^{\rho}R_j\right)}(O)]$. By Lemma 5 and submodularity, we then argue that for all $o_\ell \in T$ and at every iteration $j \leq \rho$, $\mathbb{E}\left[f_{S\cup(R_j\setminus\{o_\ell\})}(o_\ell)\right] \geq (1+\epsilon/4)\,t/k$. Thus, $o_\ell \in X_\rho$. Finally, by submodularity and the definition of $T$, we show that $f_S(T) \geq \frac{1-\epsilon/10}{2}\left((1-1/r)^{i-1}(1-\epsilon/2)\mathtt{OPT} - f(S)\right).$ $\qquad \square$

### 3.1.2 A random subset approximates the best surviving block

In the previous part of the analysis, we showed the existence of a surviving set $T$ with contribution at least $\frac{1-\epsilon/10}{2}\left((1-1/r)^{i-1}(1-\epsilon/2)\mathtt{OPT} - f(S)\right)$ to $S$. In this part, we show that the random set $R \cap X^+$, with $R \sim \mathcal{U}(X)$, is a $1/r$ approximation to any surviving set $T \subseteq X^+$ when $|X| = k$. A key component of the algorithm for this argument to hold for non-monotone functions is the final pre-processing step to restrict $X$ to $X^+$ after adding dummy elements. We use this restriction to argue that every element $a \in R \cap X^+$ must contribute a non-negative expected value to the set returned.

**Lemma 7.** *Assume* SIEVE *returns $R \cap X^+$ with $R \sim \mathcal{U}(X)$ and $|X| = k$. For any $T \subseteq X^+$, we have* $\mathbb{E}_{R\sim\mathcal{U}(X)}[f_S(R \cap X^+)] \geq f_S(T)/r$.

*Proof Sketch; full proof in Appendix D.* Since $T \subseteq X^+$, we get $\mathbb{E}[f_S(R \cap X^+)] = \mathbb{E}[f_S(R\cap T)] + \mathbb{E}[f_{S\cup(R\cap T)}((R\cap X^+)\setminus T)]$. We then use submodularity to argue that $\mathbb{E}[f_S(R\cap T)] \geq \frac{1}{r}f_S(T)$. By using submodularity and the definition of $X^+$, we then show that $\mathbb{E}[f_{S\cup(R\cap T)}((R \cap X^+) \setminus T)] \geq 0$. Combining these three pieces, we get the desired inequality. $\qquad \square$

There is a tradeoff between the contribution $f_S(T)$ of the best surviving set $T$ and the contribution of a random set $R \cap X^+$ returned in the middle of an iteration due to the thresholds $(1+\epsilon/4)t/k$ and $t/r$, which is controlled by $t$. The optimization of this tradeoff explains the $(1-\epsilon/2)/2$ term in $t$.

### 3.1.3 Proof of main lemma

*Proof of Lemma 3.* There are two cases. If SIEVE returns $R \cap X^+$ in the middle of an iteration, then by the condition to return that set, $\mathbb{E}_{R \sim \mathcal{U}(X)}[f_S(R \cap X^+)] \geq t/r = \frac{1-\epsilon/2}{2}((1-1/r)^{i-1}(1-\epsilon/2)\mathtt{OPT} - f(S))/r$. Otherwise, SIEVE returns $R \cap X^+$ with $|X| = k$. By Lemma 6, there exists $T \subseteq X_\rho$ that survives $\rho$ iterations of SIEVE s.t. $f_S(T) \geq \frac{1-\epsilon/10}{2}\left((1-1/r)^{i-1}(1-\epsilon/2)\mathtt{OPT} - f(S)\right)$. Since there are at most $\rho - 1$ iterations of SIEVE, $T$ survives each iteration and the final pre-processing. This implies that $T \subseteq X^+$ when the algorithm terminates. By Lemma 7, we then conclude that $\mathbb{E}_{R \sim \mathcal{U}(X)}[f_S(R \cap X^+)] \geq f_S(T)/r \geq \frac{1-\epsilon/10}{2r}\left((1-1/r)^{i-1}(1-\epsilon/2)\mathtt{OPT} - f(S)\right) \geq t/r$. $\qquad\square$

## 3.2 The adaptivity of SIEVE is $\mathcal{O}(\log n)$

We now observe that the number of iterations of SIEVE is $\mathcal{O}(\log n)$. This logarithmic adaptivity is due to the fact that SIEVE either returns a random set or discards a constant fraction of the surviving elements at every iteration. Similarly to Section 3.1.2, the pre-processing step to obtain $X^+$ is crucial to argue that since a random subset $R \cap X^+$ has contribution below the $t/r$ threshold and since all elements in $X^+$ have non-negative marginal contributions, there exists a large set of elements in $X^+$ with expected marginal contribution to $S \cup R$ that is below the $(1+\epsilon/4)t/k$ threshold. We defer the proof to Appendix E.

**Lemma 8.** *Let $X_j$ and $X_{j+1}$ be the surviving elements $X$ at the start and end of iteration $j$ of* SIEVE$(S, k, i, r)$. *For all $S \subseteq N$ and $r, j, \epsilon > 0$, if* SIEVE$(S, k, i, r)$ *does not terminate at iteration $j$, then $|X_{j+1}| < |X_j|/(1 + \epsilon/4)$.*

## 3.3 Main result for BLITS

**Theorem 1.** *For any constant $\epsilon > 0$,* BLITS *initialized with $r = 20\epsilon^{-1}\log_{1+\epsilon/2}(n)$ is $\mathcal{O}\left(\log^2 n\right)$-adaptive and obtains a $\frac{1-\epsilon}{2e}$ approximation.*

*Proof.* By Lemma 3, we have $\mathbb{E}[f_S(R)] \geq \frac{1-\epsilon/2}{2}\left(\left(1 - \frac{1}{r}\right)^{i-1}(1 - \epsilon/2)\mathtt{OPT} - f(S)\right)$. Thus, by Lemma 2 with $\alpha = \frac{1-\epsilon/2}{2}$ and $v^\star = (1-\epsilon/2)\mathtt{OPT}$, BLITS returns $S$ that satisfies $\mathbb{E}[f(S)] \geq \frac{1-\epsilon/2}{2e} \cdot (1-\epsilon/2)\mathtt{OPT} \geq \frac{1-\epsilon}{2e} \cdot \mathtt{OPT}$. For adaptivity, note that each iteration of SIEVE has two adaptive rounds: one for $\Delta(a, S, X)$ for all $a \in N$ and one for $\mathbb{E}_{R \sim \mathcal{U}(X)}[f_S(R \cap X^+)]$. Since $|X|$ decreases by a $1 + \epsilon/4$ fraction at every iteration of SIEVE, every call to SIEVE has at most $\log_{1+\epsilon/4}(n)$ iterations. Finally, as there are $r = 20\epsilon^{-1}\log_{1+\epsilon/4}(n)$ iterations of BLITS, the adaptivity is $\mathcal{O}\left(\log^2 n\right)$. $\qquad\square$

# 4 Experiments

Our goal in this section is to show that beyond its provable guarantees, BLITS performs well in practice across a variety of application domains. Specifically, we are interested in showing that despite the fact that the parallel running time of our algorithm is smaller by several orders of magnitude than that of any known algorithm for maximizing non-monotone submodular functions under a cardinality constraint, the quality of its solutions are consistently competitive with or superior to those of state-of-the-art algorithms for this problem. To do so, we conduct two sets of experiments where the goal is to solve the problem of $\max_{S:|S| \leq k} f(S)$ given a function $f$ that is submodular and non-monotone. In the first set of experiments, we test our algorithm on the classic max-cut objective evaluated on graphs generated by various random graph models. In the second set of experiments, we apply our algorithm to a max-cut objective on a new road network dataset, and we also benchmark it on the three objective functions and datasets used in [MBK16]. In each set of experiments, we compare the quality of solutions found by our algorithm to those found by several alternative algorithms.

## 4.1 Experiment set I: cardinality constrained max-cut on synthetic graphs

Given an undirected graph $G = (N, E)$, recall that the cut induced by a set of nodes $S \subseteq N$ denoted $C(S)$ is the set of edges that have one end point in $S$ and another in $N \setminus S$. The cut function

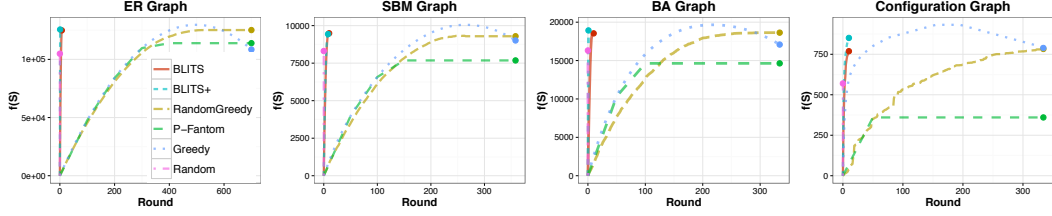

Figure 1: *Experiments Set 1: Random Graphs.* Performance of BLITS (red) and BLITS+ (blue) versus RANDOMGREEDY (yellow), P-FANTOM (green), GREEDY (dark blue), and RANDOM (purple).

$f(S) = |C(S)|$ is a quintessential example of a non-monotone submodular function. To study the performance of our algorithm on different cut functions, we use four well-studied random graph models that yield cut functions with different properties. For each of these graphs, we run the algorithms from Section 4.3 to solve $\max_{S:|S| \leq k} |C(S)|$ for different $k$:

- **Erdős Rényi.** We construct a $G(n, p)$ graph with $n = 1000$ nodes and $p = 1/2$. We set $k = 700$. Since each node's degree is drawn from a Binomial distribution, many nodes will have a similar marginal contribution to the cut function, and a random set $S$ may perform well.

- **Stochastic block model.** We construct an SBM graph with 7 disconnected clusters of 30 to 120 nodes and a high ($p = 0.8$) probability of an edge within each cluster. We set $k = 360$. Unlike for $G(n, p)$, here we expect a set $S$ to achieve high value only by covering all of the clusters.

- **Barbási-Albert.** We create a graph with $n = 500$ and $m = 100$ edges added per iteration. We set $k = 333$. We expect that a relatively small number of nodes will have high degree in this model, so a set $S$ consisting of these nodes will have much greater value than a random set.

- **Configuration model.** We generate a configuration model graph with $n = 500$, a power law degree distribution with exponent 2. We set $k = 333$. Although configuration model graphs are similar to Barbási-Albert graphs, their high degree nodes are not connected to each other, and thus greedily adding these high degree nodes to $S$ is a good heuristic.

### 4.2 Experiment set II: performance benchmarks on real data

To measure the performance of BLITS on real data, we use it to optimize four different objective functions, each on a different dataset. Specifically, we consider a traffic monitoring application as well as three additional applications introduced and experimented with in [MBK16]: image summarization, movie recommendation, and revenue maximization. We note that while these applications are sometimes modeled with monotone objectives, there are many advantages to using non-monotone objectives (see [MBK16]). We briefly describe these objective functions and data here and provide additional details in Appendix F.

- **Traffic monitoring.** Consider an application where a government has a budget to build a fixed set of monitoring locations to monitor the traffic that enters or exits a region via its transportation network. Here, the goal is not to monitor traffic circulating within the network, but rather to choose a set of locations (or nodes) such that the volume of traffic entering or exiting via this set is maximal. To accomplish this, we optimize a cut function defined on the weighted transportation network. More precisely, we seek to solve $\max_{S:|S| \leq k} f(S)$, where $f(S)$ is the sum of weighted edges (e.g. traffic counts between two points) that have one end point in $S$ and another in $N \setminus S$. To conduct an experiment for this application, we reconstruct California's highway transportation network using data from the CalTrans PeMS system [Cal], which provides real-time traffic counts at over 40,000 locations on California's highways. Appendix F.1 details this network reconstruction. The result is a directed network in which nodes are locations along each direction of travel on each highway and edges are

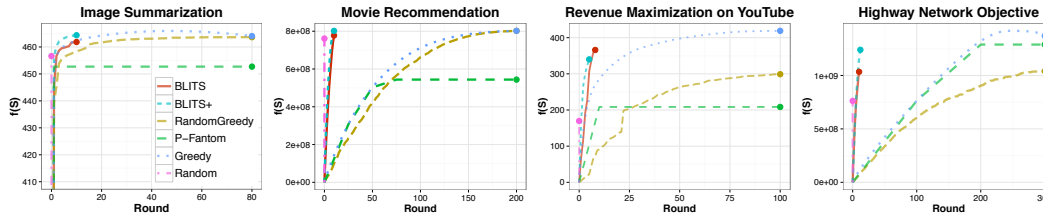

Figure 2: *Experiments Set 2: Real Data.* Performance of BLITS (red) and BLITS+ (blue) versus RANDOMGREEDY (yellow), P-FANTOM (green), GREEDY (dark blue), and RANDOM (purple).

the total count of vehicles that passed between adjacent locations in April, 2018. We set $k = 300$ for this experiment.

- **Image summarization.** Here we must select a subset to represent a large, diverse set of images. This experiment uses 500 randomly chosen images from the *10K* Tiny Images dataset [KH09] with $k = 80$. We measure how well an image represents another by their cosine similarity.

- **Movie recommendation.** Here our goal is to recommend a diverse short list $S$ of movies for a user based on her ratings of movies she has already seen. We conduct this experiment on a randomly selected subset of 500 movies from the MovieLens dataset [HK15] of 1 million ratings by 6000 users on 4000 movies with $k = 200$. Following [MBK16], we define the similarity of one movie to another as the inner product of their raw movie ratings vectors.

- **Revenue maximization.** Here we choose a subset of $k = 100$ users in a social network to receive a product for free in exchange for advertising it to their network neighbors, and the goal is to choose users in a manner that maximizes revenue. We conduct this experiment on 25 randomly selected communities ($\sim$1000 nodes) from the 5000 largest communities in the YouTube social network [FHK15], and we randomly assign edge weights from $\mathcal{U}(0, 1)$.

## 4.3 Algorithms

We implement a version of BLITS exactly as described in this paper as well as a slightly modified heuristic, BLITS+. The only difference is that whenever a round of samples has marginal value exceeding the threshold, BLITS+ adds the highest marginal value sample to its solution instead of a randomly chosen sample. BLITS+ does not have any approximation guarantees but slightly outperforms BLITS in practice. We compare these algorithms to several benchmarks:

- **RandomGreedy**. This algorithm adds an element chosen u.a.r. from the $k$ elements with the greatest marginal contribution to $f(S)$ at each round. It is a $1/e$ approximation for non-monotone objectives and terminates in $k$ adaptive rounds [BFNS14].

- **P-Fantom**. P-FANTOM is a parallelized version of the FANTOM algorithm in [MBK16]. FANTOM is the current state-of-the-art algorithm for non-monotone submodular objectives, and its main advantage is that it can maximize a non-monotone submodular function subject to a variety of intersecting constraints that are far more general than cardinality constraints. The parallel version, P-FANTOM, requires $\mathcal{O}(k)$ rounds and gives a $1/6 - \epsilon$ approximation.

We also compare our algorithm to two reasonable heuristics:

- **Greedy.** GREEDY iteratively adds the element with the greatest marginal contribution at each round. It is $k$-adaptive and may perform arbitrarily poorly for non-monotone functions.

- **Random.** This algorithm merely returns a randomly chosen set of $k$ elements. It performs arbitrarily poorly in the worst case but requires 0 adaptive rounds.

## 4.4 Experimental results

For each experiment, we analyze the value of the algorithms' solutions over successive rounds (Fig. 1 and 2). The results support four conclusions. First, BLITS and/or BLITS+ nearly always found solutions whose value matched or exceeded those of FANTOM and RANDOMGREEDY— the two alternatives we consider that offer approximation guarantees for non-monotone objectives. This also implies that BLITS found solutions with value far exceeding its own approximation guarantee, which is less than that of RANDOMGREEDY. Second, our algorithms also performed well against the top-performing algorithm — GREEDY. Note that GREEDY's solutions decrease in value after some number of rounds, as GREEDY continues to add the element with the highest marginal contribution each round even when only negative elements remain. While BLITS's solutions were slightly eclipsed by the *maximum* value found by GREEDY in five of the eight experiments, our algorithms matched GREEDY on Erdős Rényi graphs, image summarization, and movie recommendation. Third, our algorithms achieved these high values despite the fact that their solutions $S$ contained $\sim$10-15% fewer than $k$ elements, as they removed negative elements before adding blocks to $S$ at each round. This means that they could have actually achieved even higher values in each experiment if we had allowed them to run until $|S| = k$ elements. Finally, we note that BLITS achieved this performance in many fewer adaptive rounds than alternative algorithms. Here, it is also worth noting that for all experiments, we initialized BLITS to use only 30 samples of size $k/r$ per round — far fewer than the theoretical requirement necessary to fulfill its approximation guarantee. We therefore conclude that in practice, BLITS 's superior adaptivity does not come at a high price in terms of sample complexity.

## Acknowledgments

This research was supported by a Google PhD Fellowship, NSF grant CAREER CCF 1452961, NSF CCF 1301976, BSF grant 2014389, NSF USICCS proposal 1540428, a Google Research award, and a Facebook research award.

## Footnotes

[1]To date, the best upper and lower bounds are [BFNS14] and [GV11] respectively for non-monotone submodular maximization under a cardinality constraint.

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
