[Reviews · NeurIPS 2018]

Reviewer 1



This paper focuses on non-monotone submodular maximization problem. In this problem we are given a normalized non-negative submodular function ‘f’ and the goal is to find a subset of size at most k maximizing the value of ‘f’. Submodular maximization problem is a well-known and well-studied problem in different settings, e.g., offline, streaming, online, map reduce,… . It has been also studied under different assumption, monotone, cardinality constraint, matroid constraint, normalized, non-negative,… . The greedy algorithm achieves a 1/e-approximate solution in O(k) adaptive rounds of accessing the data. This work presents a 1/2e-approximation algorithm that need poly(log n) adaptive rounds, which makes it more suitable for massive data sets. In my point of view the paper is well-written and authors have explained the key ideas of the algorithm and the analysis nicely. Both the problem that they consider and the algorithm that they propose are interesting. The author claim that they achieve “Exponentially fewer Iterations”, which is not fully correct. An exponential improvement results in a poly(log k) iterations and not poly(log n). For some cases these two are equivalent, both not for all the cases. Therefore, I think its better to change the title. They also propose a subroutine called SIEVE, which is very similar to the name of a very famous algorithm for monotone submodular maximization in streaming setting. At first, I thought the authors are using that algorithm. I suggest to change the name to avoid confusions. Also in case they have enough space, it might be better to explain the sampling process more. In general I think the paper is in a good shape and the result is nice.

Reviewer 2



The authors consider the parallel complexity of non-monotone submodular maximization subject to a cardinality constraint. They develop an algorithm for this problem and they are able to show it has a (1 / 2e)-factor optimization for this problem in polylog(n) parallel rounds. They experimentally observe that their algorithm is able to achieve a good approximation in far fewer rounds. The theoretical results of this paper are very strong. It is typically difficult to obtain good approximation guarantees for non-monotone functions. The paper is well written. The author's algorithm is a tweak of the algorithmic framework proposed in [BS18a, BS18b, BRS18]. Some of the proof ideas are similar to BRS18. The authors should better differentiate the proof for this problem by discussing the difficulties unique to this problem. One important experiment missing is running a real parallel or distributed implementation of the algorithm and observing how well the algorithm parallelizes in practice. How does the performance improve with the number of processors and how does this compare to parallel implementations of other algorithms such as the greedy algorithm? The guess-and-check idea for the algorithm is related to the algorithm in the following work: Badanidiyuru, Ashwinkumar, et al. "Streaming submodular maximization: Massive data summarization on the fly." Proceedings of the 20th ACM SIGKDD international conference on Knowledge discovery and data mining. ACM, 2014. The authors should cite this work. One improvement to the psedocode: if E_{R ~ U(X)}[...] <= ..., then sample R ~ U(X) and return R \cap X^+ As written, it was somewhat confusing at first with R as both the variable under the expectation and the random sample. The authors should clearly write the value of k used in the experiments, as this defines the number of iterations for the competing algorithms. *** Post-Rebuttal *** Thank you for your response. While I agree that the experiments provided bring insights that a real parallel implementation would not, I still think a real parallel implementation would be useful. I think simply seeing how the running time decreases as the number of CPUs increases would be insightful. Also it seems to me that the RandomGreedy algorithm has a natural Map/Reduce style parallel algorithm that could be used for comparison.

Reviewer 3



In this paper, the authors study the problem of non-monotone submodular maximization and they proposed an algorithm with slightly worse guarantee but significant less adaptivity. Here are a few detailed comments, 1. The proposed algorithm is paralleled, but I did find the number of machines required. Can the authors make it clearer? It is critical, because one could easily come up with a paralleled exhaustive search algorithm that is 1-adaptive, always finds the true optima but require exponential number of CPU. Also, around line 115-117, how many samples is needed for calculating \Delta? 2. In the Figure 1 and 2, the x-axis is the number of rounds. But different algorithms have different time complexity per round. Can the author make other plots with respect to actual running time and total running time of all paralleled machine?